POINT OF VIEW

# How should novelty be valued in science?

**Abstract** Scientists are under increasing pressure to do "novel" research. Here I explore whether there are risks to overemphasizing novelty when deciding what constitutes good science. I review studies from the philosophy of science to help understand how important an explicit emphasis on novelty might be for scientific progress. I also review studies from the sociology of science to anticipate how emphasizing novelty might impact the structure and function of the scientific community. I conclude that placing too much value on novelty could have counterproductive effects on both the rate of progress in science and the organization of the scientific community. I finish by recommending that our current emphasis on novelty be replaced by a renewed emphasis on predictive power as a characteristic of good science.

**BARAK A COHEN**[*]

**\*For correspondence:** cohen@wustl.edu

## Introduction

*"(T)he primary **novelty** of this work is the ability to make a prediction about drug sensitivity. Reviewers felt that the predictive ability would be very hard to generalize, however, reducing the impact of this **novel** feature. This concern about **novelty**... was the driving factor in this decision."*

-excerpt from a rejection letter received by the author

A mere 48 years separates the discovery of the double-helix structure of DNA (*Watson and Crick, 1953*) from the announcements that the human genome had been sequenced (*Lander et al., 2001*; *Venter et al., 2001*). The pace and regularity with which important discoveries have been made in molecular biology is remarkable. Molecular biologists have had an uncanny knack of homing in on the small irregularities that lead to large breakthroughs. It was irregularly colored ears of corn that revealed the existence of mobile genetic elements known as transposons (*McClintock, 1950*). Many of the most important regulators of human development first surfaced as mutations that slightly alter the rows of bristles on the undersides of fruit fly larvae (*Nüsslein-Volhard and Wieschaus, 1980*). Scientists studying tiny roundworms that age in odd ways helped uncover micro RNAs (*Lee et al., 1993*; *Wightman et al., 1993*), which are now thought to regulate a large fraction of human genes. Again and again molecular biologists have seized on these sorts of minutiae to gain enormous insight into the inner workings of cells. Looking back over the last 60 years one feels a great sense of pride in being part of a tradition that is undoubtedly one of the most productive in the history of science.

Given the winning formula molecular biologists appear to have hit on, it is interesting that there are large changes occurring in our community. As the size of the molecular biology community continues to grow, competition for limited funding has become much more intense. With the completion of the human genome has come immense pressure to "translate" basic research findings into new treatments for disease. In the United States our institutional leaders at the National Institutes of Health (NIH) openly worry about data showing that the rate of discovery in the biomedical sciences no longer reflects the size of their investments (*Cook et al., 2015*; *Fortin and Currie, 2013*; *Gallo et al., 2014*; *Lauer et al., 2015*; *Doyle et al., 2015*). Undoubtedly these pressures influence the trajectories of research programs. What we do not know yet is how these pressures impact the overall productivity of our community.

One manifestation of these changes is an increasing emphasis on "novelty" in science. Our scientific establishment – through our funding agencies, review panels and editorial boards – are clearly putting a higher and higher premium on research that is deemed novel. Research programs that lack a "high degree" of novelty struggle for support and "incremental" findings are relegated to publication in second- and third-tier journals. NIH grant proposals now have an "Innovation" section where investigators must explicitly list the attributes of their research that make it novel. While funding agencies seek novelty in their grant portfolios, they are also increasingly looking for "feasibility" as resources become scarce, and this appears to put novel research programs at a disadvantage (*Alberts et al., 2014*). As investigators struggle to walk a nearly impossible line between feasibility and novelty, the definition of novelty itself becomes blurred. Novelty can now mean anything from demonstrating a well-established phenomenon in a new system to testing a hypothesis with no precedent in the literature. Even though we cannot strictly define what is and is not novel, the message is still clear; novelty equates with good research.

Perhaps this emphasis on novelty is not really new at all, but only a codifying of something we already value implicitly. Even so, we should consider the effects that an explicit emphasis on novelty might have on the properties of scientific research that have made molecular biology so successful. These properties include our system of peer review, our scientific standards of proof and falsification, and the organization of the scientific community. Increasing the value we place on novelty will likely affect each of these factors.

## It appears then that nothing in the ideas of Popper or Kuhn particularly values novelty for its own sake.

### Lessons from the philosophy of science

For working scientists Karl Popper is almost certainly the most influential philosopher of science. Most of us at least pay lip service to Popper's philosophy when we recite the mantra that hypotheses can never be proved, only disproved. For many scientists the distinction between what is disprovable and what is not demarcates the line between what is and is not science, an idea taken directly from Popper's writings. According to Popper, scientists propose new hypotheses about how the world works, and any hypotheses that are subsequently falsified by empirical observation are relegated to the scrap heap (*Popper, 1963*). This framework of hypothesis generation and refutation is widely accepted by scientists.

What is less well appreciated is how utterly Popper rejected the notion of confirmation. Popper was adamant that the survival of a hypothesis in the face of empirical challenge says nothing about its validity, only that that the hypothesis has yet to be falsified. However, Popper's strict adherence to this idea became difficult to defend and, to be practical, most scientists do allow that empirical evidence can either support or falsify a hypothesis.

What if anything can we infer about the value of novelty from Popper's ideas on hypotheses and falsification? Because Popper believed that hypotheses can never be proved, he stressed that hypotheses must be subjected to repeated testing, even after they have survived several empirical challenges. In this sense he valued follow-through over novelty. However, because Popper believed that "good tests kill flawed theories", new tests must be more than trivial variations of previous experiments. The philosopher Imre Lakatos argued that good research programs are "progressive" (*Lakatos, 1970*), and that scientists should constantly seek to expand their hypotheses into new areas of observation. Today, however, review panels are likely to tag progressive research programs as lacking in novelty because the scientists who pursue these programs seek to expand old hypotheses into new realms, rather than develop new hypotheses altogether. This is misguided. Scientists following progressive research programs require ingenuity and creativity to devise the tests that expand the reach of their hypotheses beyond the obvious. According to Popper the novelty of a new hypothesis is beside the point, unless and until the hypothesis it is meant to replace is falsified.

Thomas Kuhn, a contemporary of Popper, was in many ways Popper's opposite. Kuhn emphasized the importance of "paradigms", coherent collections of claims, methodologies, and teaching practices that govern scientific

inquiry. In his hugely influential book *The Structure of Scientific Revolutions* he explains that the purpose of a paradigm is to provide a guide for investigating the right questions (*Kuhn and Hacking, 2012*). Here Kuhn's philosophy sharply contrasts with Popper's. While Popper advocated abandoning a theory the moment it was falsified, Kuhn emphasized that paradigms can tolerate a good deal of "anomalies" and still remain valid. The flexibility of paradigms allows scientists to continue working in a productive framework long after falsification would have dictated a change. If scientists had to drop their paradigms every time they encountered a problem then nothing would ever get done. Only a critical mass of anomalies requires a "paradigm shift".

It appears then that nothing in the ideas of Popper or Kuhn particularly values novelty for its own sake. Both Popper and Kuhn emphasized the need for scientists to stick doggedly with their hypotheses, Popper because hypotheses must be challenged continually no matter how often they have been confirmed, and Kuhn because only a critical mass of anomalies can force a paradigm shift. Ironically, over time the effect of Kuhn's book has been to weaken scientists' belief in their paradigms. Many investigators now actively search for paradigm shifts. This conflicts with Kuhn's description of progress in which scientists cling tightly to their paradigms, giving them up only grudgingly after the weight of anomalous results renders the paradigm unsupportable. Despite their differences, novelty seeking is not a key component in the philosophies of either Popper or Kuhn.

Many scientists have a visceral reaction to philosophies that cast them as mechanically pursuing their hypotheses. Kuhn in particular was attacked for seeming to endorse a grinding and boring type of science, and he did not help his case by referring to work done in the context of a paradigm as "normal" science.

But we need not explicitly value novelty to keep science from being a dull grind. Peter Godfrey-Smith writes that Popper painted an appealing picture of scientists as "hard-headed cowboys, out on the range, with a Stradivarius tucked in their saddlebags" (*Godfrey-Smith, 2003*). Hard-headed because they must have the determination to stick with their hypotheses, and packing a Stradivarius because they need inspiration when devising tests that expand their hypotheses into new realms. Kuhn too seemed in awe of the ability of normal science to hone in on "miniscule" findings that end up revealing deep truths about the world. Think of the little tails on the electron micrographs of the RNA:DNA hybrids that revealed the phenomenon of intron splicing (*Berget et al., 1977*), or the examples given at the start of this article. While normal science might seem a derogatory term for what most investigators do, Kuhn saw it as requiring imagination.

Even still, as working scientists we know that much of day-to-day science involves painstaking and often repetitive work. Science succeeds because powerful social incentives help us push through the less glamorous aspects of research. Godfrey-Smith writes that the most significant reactions to the philosophies of both Popper and Kuhn emphasized the importance of social forces in science. For example, in his later writings Popper struggled with the question of exactly when an observation counts as a refutation. His solution was to shift from describing the proper methodologies of science to describing the proper social behavior of scientists. For Kuhn, paradigms highlighted the importance of the social aspects of science, including the indoctrination of students and the collective adherence to particular claims among investigators working under the same paradigm. In the next section I discuss how the increasing emphasis on novelty might influence the social structure of science.

## Lessons from the sociology of science

An important question for sociologists of science – and also for scientists and funding agencies – is: What distribution of people across rival research programs is best for science? The immediate impact of emphasizing novelty might be to distribute researchers over the widest possible range of research programs, as each investigator seeks to maximize the novelty of their own research program. This might seem an efficient way of exploring the widest possible range of theories but such a distribution also raises problems. Kuhn wrote extensively of the necessity of having large groups of researchers organized around a particular set of theories. Placing too much emphasis on novelty may result in a distribution of effort that is too diffuse to enable efficient progress. But scientists consider an array of incentives besides novelty when choosing their research programs.

Robert Merton laid the foundations of the sociology of science with his discussion of reward systems in science (*Merton, 1957*).

Merton argued that recognition is the main form of reward in science. In particular the "priority rule", which awards the most recognition to the first investigator to support a hypothesis, is an especially powerful incentive in science. To support his idea Merton showed that the history of science is chock full of disputes over priority (for example, Isaac Newton battled Gottfried Leibniz over priority for the invention of calculus (**Hall, 1980**)). One benefit of an incentive system that rewards priority is that it encourages original thought and novel lines of investigation. One might argue that this means that novelty seeking is already baked directly into the social fabric of science.

> **Hull viewed the success of science as a result of a delicate balance between competition and cooperation, creativity and skepticism, trust and doubt, and open-mindedness and dogmatism. Placing too much emphasis on novelty could upset this equilibrium in ways that are not optimal for scientific progress.**

Some sociologists argue that the priority incentive coupled with the individual quest for credit is what produces good outcomes in the scientific community. These authors envision something like the "invisible hand" that guides free market capitalism in Adam Smith's *Wealth of Nations* (**Smith, 2000**). Scientists must balance risk versus reward when choosing between competing hypotheses to explore. The priority incentive prevents all investigators from working on the hypothesis with the highest probability of success. The argument is that credit is a pie of fixed size that can be shared either equally (**Kitcher, 1990**) or unequally (**Strevens, 2003**), but only by investigators who work on the winning hypothesis. When too many scientists work on the same hypothesis there is an incentive to work on novel hypotheses, even ones where the chance of success might be smaller, but where

the share of credit would be larger (**Laudan, 1977**). In this way the priority rule balances cooperation and competition between scientists, and divides individual effort between different research programs.

David Hull argued that science is particularly good at portioning effort in a way that maximizes good outcomes for the community (**Hull, 1988**). Hull agreed with Merton that the priority rule helps to maintain a balance between cooperation and competition in science. However, he also recognized the importance of the rivalries between scientists that encourage investigators to check the validity of their competitors' work, especially results they may want to use in their own research. This checking, along with the priority rule, helps to maintain a balance between creativity and skepticism, which Hull believed was an essential feature of science. Scientists can become overly attached to their ideas, and most are reluctant to kill their pet theories, especially theories with creative panache. To counterbalance this tendency science relies on the incentive rival scientists have to vigorously check work that may be useful to them, or results that challenge their own dogma.

Hull might have been wary about introducing an explicit incentive for novelty into the scientific community. For one thing, along with most other sociologists of science, he thought that the priority incentive already provided a powerful motivation for scientists to test novel theories. But more than others Hull viewed the success of science as a result of a delicate balance between competition and cooperation, creativity and skepticism, trust and doubt, and open-mindedness and dogmatism. Placing too much emphasis on novelty could upset this equilibrium in ways that are not optimal for scientific progress.

In particular, an explicit emphasis on novelty might perturb the balance between the incentive for scientists to check their rivals' theories and the priority rule. The priority rule provides a powerful incentive for scientists to publish their work quickly. This is good for the community because new ideas get disseminated rapidly, where they can be incorporated into other research programs. However, there is an equally powerful incentive to be correct when publishing because scientists know that other investigators who want to build on their results are likely to uncover any mistakes that make it into print. If we value novelty too much then scientists will be incentivized to publish too quickly, without

imposing the rigor they might normally demand of themselves. Progress would slow to a crawl as other scientists waste time trying to build on flawed results.

Indeed, some in the scientific establishment have already warned of a "crisis in reproducibility" (*Errington et al., 2014*; *Baker, 2016*). Not surprisingly this crisis follows an explosion in papers reporting weak claims of novelty (*Henikoff and Levis, 1991*; *Friedman and Karlsson, 1997*). Others have argued that the reward system in modern molecular biology incentivizes statistically underpowered research designs (*Higginson and Munafò, 2016*). To counteract this trend some of the leaders in our field now advocate funding centralized efforts to validate published studies (*Collins and Tabak, 2014*). This suggests that priority and checking have become unbalanced in the general scientific community. Those leaders advocating for centralized checking efforts might do well to ask themselves what role their emphasis on novelty has played in precipitating this so-called crisis.

Another consequence of emphasizing novelty might be to increase the tenacity with which scientists attack their rivals' hypotheses. Novel results are particularly likely to be attacked, in part because scientists who can lay claim to novelty enjoy so many advantages over other scientists. Rival scientists are thus incentivized to use anomalous results to discredit novel hypotheses. This is unfortunate because as Kuhn emphasized, hypotheses must be allowed to tolerate some anomalous results before they are discarded, otherwise the community cannot exploit the utility of working models. Ironically, novel research programs have a very difficult time surviving when novelty is so highly coveted.

> **Perhaps our obsession with novelty is a sort of communal nostalgia for the good old days, when important foundational discoveries came fast and furious.**

An emphasis on novelty could also break the cohesion between scientists working within research programs. Cooperation is essential to scientific progress, and this cooperation is balanced by competition from investigators who are willing to challenge rival theories. If scientists must maximize the novelty of their research then they are more likely to pursue avenues as different as possible from their colleagues. We risk producing a community in which no single paradigm has the critical mass of supporters required to function effectively. This is a serious problem because current paradigms, imperfect though they might be, often have great utility, even though they may eventually be revised or even discarded.

## Conclusions

When an area of science experiences rapid advancement over a short interval of time it may be followed by a period in which novel discoveries are harder to come by. After Mendeleyev articulated the concept of the periodic table there was an exciting period in which novel elements were rapidly discovered. As time passed it became more and more difficult to isolate the remaining elements. Perhaps molecular biology is also in a lull after a period of virtually unprecedented achievement. Almost 50 years ago Gunther Stent argued that there were no new principles left to discover in molecular biology (*Stent, 1969*). All that scientists could look forward to would be the tedious grind of filling in details. These sorts of pronouncements have a way of being undone by events. For example, Stent's prediction came before the discovery of splicing, reverse transcription, and micro RNAs. Even so, it may well be true that most of the foundational principles of molecular biology have already been discovered. Perhaps our obsession with novelty is a sort of communal nostalgia for the good old days, when important foundational discoveries came fast and furious.

It might also be that our desire to reward novelty stems from the frustration that research in molecular biology is not "translating" into new practical applications as fast as some might wish. The endless overpromising of novel therapeutics from our institutional leaders only makes this matter worse. Why don't discoveries in molecular biology translate more quickly into practical applications? Is it because we are missing large chunks of basic theory? Probably not, and those who go searching for novelty and paradigm shifts are likely to be disappointed.

Instead, we face a very different set of problems. While our models are generally quite good at explaining the basic mechanisms underlying molecular biology, it is also the case that most

of our models lack a quantitative formulation. Even when we know the underlying molecular mechanisms at work in a given system or process, in most cases we lack the ability to make quantitative predictions about the effects that specific perturbations will have on that system or process. We have a mountain of facts about how transcription initiates and beautiful cartoon models of this process, but we cannot predict the effects that genetic variants will have on transcription rates, whether these variants reside in *cis*-acting DNA sequences or in *trans*-acting protein factors. We know the identities of virtually all the proteins involved in apoptosis, and which of their post-translational modifications are pro- or anti-apoptotic. Yet we cannot use quantitative measures of the levels of these proteins in any cell type to make an accurate prediction of whether that cell will die or not. We understand the principles that drive peptide sequences to fold into secondary and tertiary structures, yet we cannot predict the shape any given amino acid sequence will adopt. Seen through the lens of predictive power, it is clear that the vast majority of models in molecular biology are inadequate for solving real world problems.

If we want to solve important practical problems then progressive research programs that expand and refine the predictive power of existing models are at least as important as research programs focused on novel hypotheses. One suggestion would be to replace the current emphasis on novelty with an emphasis on predictive power, particularly quantitative predictions. Research that results in models that reliably and quantitatively predict the outcomes of genetic, biochemical, or pharmacological perturbations should be valued highly, and rewarded, regardless of whether such models invoke novel phenomena.

The increasing emphasis placed on novelty brings significant dangers. As it becomes more and more important for scientists to be "the first to demonstrate" some claim, the influence of the priority rule will increase and more scientists will feel pressure to sacrifice rigor for speed of publication. We are also likely to see an increase in distasteful disputes over priority. The cohesion between competing groups may also be in jeopardy as the drive for novelty distorts the balance between competition and cooperation that has characterized the success of molecular biology over the past several decades.

Science as we practice it today is a relatively recent development. Our system of peer review, the priority rule, and the organization of scientists into cooperative social demes that compete against other groups of scientists all trace their origin to decisions made by the Royal Society in the late 1600s. For most of history humans acquired knowledge outside of what we would recognize as a scientific framework. It would be unwise to assume that science is a permanent feature of our society or that it can withstand deep structural changes and remain an efficient engine of discovery. The explicit value we now place on novelty in molecular biology is a change we should approach with caution if we are to safeguard the essential features of science that have made our field so successful.

## Acknowledgements

I thank Rob Mitra, Mark Johnston, Siqi Zhao, Max Staller, Michael White, Zach Pincus, and Dana King for critical readings of the manuscripts and engaging discussions.

**Barak A Cohen** is in the Edison Family Center for Genome Sciences and Systems Biology and Department of Genetics, Washington University School of Medicine, Saint Louis, United States

ⓘ http://orcid.org/0000-0002-3350-2715

*Competing interests:* The author declares that no competing interests exist.

*Author contributions:* BAC, Conceptualization, Writing—original draft, Writing—review and editing

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
