## [Decision Letter]

Thank you for submitting your manuscript "How should novelty be valued in science?" to *eLife* for consideration as a Feature Article. Your manuscript has been reviewed by two peer reviewers and the *eLife* Features Editor (Peter Rodgers). The following individuals involved in review of your submission have agreed to reveal their identity: Yitzhak Pilpel (Reviewer #1) and Angela H DePace (Reviewer #2).

The reviewers have discussed the reviews with one another and the Features Editor has drafted this decision to help you prepare a revised submission. Most of the major revisions requested are optional (we feel the article would be improved if you addressed them, but it is not essential that you do).

Summary:

The paper is an impressive scholarly work. It is broad, deep and methodological. It is very well written (though perhaps could be shortened). It studies the value of novelty in science through several angles, including philosophy of science (the excellent survey and comparison of Popper's vs. Kuhn's teachings as well as other less well-known thinkers is used here very effectively to deliver the notion that both falsification as well as paradigm establishment and shifting require more than purely "novelty-science"); it considers very effectively social and cultural aspects of science (the role of fame and recognition in the process, competition etc.); it touches upon the emotional aspects of doing science, and it very effectively also touches upon science organization and policy aspects such as in funding and granting of research projects (where the call for funding, not only individualistic research is refreshing and, in a way novel, in the current atmosphere).

Major revisions:

1) The solution presented at the end (to focus on quantitative prediction as a gauge of novelty) is only one of many possible solutions, and it would be good if the author could discuss other possible solutions, although we should not insist on this.

I would argue that another solution would be including some description of the sociology of science in graduate and undergraduate education, such that the value of novelty and reproducibility/extension at the community level are more clear to people. Right now we almost exclusively lift up isolated geniuses as scientific heroes; is it no wonder that everyone chases some paradigm shift of their own? I'm sure there are other solutions as well.

2) A common complaint I hear is that the competitive nature of modern science means that authors often over-sell their findings in papers in order make them seem more novel than they really are. Again, it would be good if the author could briefly discuss this phenomenon.

3) In addition to the relationship between novelty and philosophical and sociological factors it would be good to discuss how competition for funding and jobs seems to be reducing novelty – as outlined, for example, in the following passage from Alberts et al. 2014. Rescuing US Biomedical Research from its Systemic Flaws. PNAS 111:5773-5777:

"Competition in pursuit of experimental objectives has always been a part of the scientific enterprise, and it can have positive effects. However, hypercompetition for the resources and positions that are required to conduct science suppresses the creativity, cooperation, risk-taking, and original thinking required to make fundamental discoveries.

Now that the percentage of NIH grant applications that can be funded has fallen from around 30% into the low teens, biomedical scientists are spending far too much of their time writing and revising grant applications and far too little thinking about science and conducting experiments. The low success rates have induced conservative, short-term thinking in applicants, reviewers, and funders. The system now favors those who can guarantee results rather than those with potentially path-breaking ideas that, by definition, cannot promise success. Young investigators are discouraged from departing too far from their postdoctoral work, when they should instead be posing new questions and inventing new approaches. Seasoned investigators are inclined to stick to their tried-and-true formulas for success rather than explore new fields.

One manifestation of this shift to short-term thinking is the inflated value that is now accorded to studies that claim a close link to medical practice […]".

It would be good to discuss these matters (in just a paragraph or two) in part 1 or part 4 of the article, but this is not essential.

4) I would consider swapping the order of sections 2 and 3. Section 3 is the stronger of the two, in my opinion, and describes one ideal version of how the scientific community functions that many of us are familiar with, at least in the abstract. It thus may serve as more of a common starting point. (Although it may be worth noting that some aspects of this ideal might not serve us well either. For example it is highly individualistic and competitive in its framing; the same goals of novelty seeking and cross-checking might be achieved by other more collaborative social structures). The segue to section 2 can then be that novelty-seeking is a requirement of the social structure described in the previous section, as is independently validating or extending results in new areas. Both of these activities can be accommodated in the philosophical frameworks presented, but there is a clear second-tier status assigned to validating or extending results in some of them. Thus the dominant influence of Kuhn's work can be seen to be somewhat destructive in the overall goals of science. (Everyone constantly seeking poorly-defined paradigm shifts isn't necessarily productive).

---

## [Author Response]

As directed in the decision letter I have addressed some, but not all, of the major points as the letter indicated that addressing these points was optional.

*Major revisions:*

*1) The solution presented at the end (to focus on quantitative prediction as a gauge of novelty) is only one of many possible solutions, and it would be good if the author could discuss other possible solutions, although we should not insist on this.*

*I would argue that another solution would be including some description of the sociology of science in graduate and undergraduate education, such that the value of novelty and reproducibility/extension at the community level are more clear to people. Right now we almost exclusively lift up isolated geniuses as scientific heroes; is it no wonder that everyone chases some paradigm shift of their own? I'm sure there are other solutions as well.*

*2) A common complaint I hear is that the competitive nature of modern science means that authors often over-sell their findings in papers in order make them seem more novel than they really are. Again, it would be good if the author could briefly discuss this phenomenon.*

This point is addressed in the ninth paragraph of the section “Lessons from the sociology of science”. I cite to papers documenting the exponential rise in claims to novelty.

3) In addition to the relationship between novelty and philosophical and sociological factors it would be good to discuss how competition for funding and jobs seems to be reducing novelty – as outlined, for example, in the following passage from Alberts et al. 2014. Rescuing US Biomedical Research from its Systemic Flaws. PNAS 111:5773-5777:

*"Competition in pursuit of experimental objectives has always been a part of the scientific enterprise, and it can have positive effects. However, hypercompetition for the resources and positions that are required to conduct science suppresses the creativity, cooperation, risk-taking, and original thinking required to make fundamental discoveries.*

*Now that the percentage of NIH grant applications that can be funded has fallen from around 30% into the low teens, biomedical scientists are spending far too much of their time writing and revising grant applications and far too little thinking about science and conducting experiments. The low success rates have induced conservative, short-term thinking in applicants, reviewers, and funders. The system now favors those who can guarantee results rather than those with potentially path-breaking ideas that, by definition, cannot promise success. Young investigators are discouraged from departing too far from their postdoctoral work, when they should instead be posing new questions and inventing new approaches. Seasoned investigators are inclined to stick to their tried-and-true formulas for success rather than explore new fields.*

*One manifestation of this shift to short-term thinking is the inflated value that is now accorded to studies that claim a close link to medical practice […]".*

*It would be good to discuss these matters (in just a paragraph or two) in part 1 or part 4 of the article, but this is not essential.*

I now address this point in the Introduction (fourth paragraph) and cite the Alberts et al. (2014) paper.